# Green Agendas and White Markets: The Coloniality of Agroecology in Senegal

Franziska Marfurt [1,*], Tobias Haller [2] and Patrick Bottazzi [1]

1   Institute of Geography, University of Bern, 3012 Bern, Switzerland; patrick.bottazzi@unibe.ch
2   Institute of Social Anthropology, University of Bern, 3012 Bern, Switzerland; tobias.haller@unibe.ch
*   Correspondence: franziska.marfurt@unibe.ch

**Abstract:** Development actors in West Africa have been promoting agroecological farming as a solution to combat climate change and to create more sovereign food systems that enhance the autonomy of local smallholders. However, there is a lack of empirical evidence regarding the actual implementation of such programs and their potential to empower smallholders, especially in the West African region. Drawing on co-produced knowledge from anthropological fieldwork in Western Senegal, the case study of an alternative food network explores the interlinkages between the promotion of agroecology, anti-migration policies, and unequal power and market relations. Informed by decolonial political ecologies, the analysis reveals different layers of coloniality which complicate embodied effects on horticultural smallholders. The authors conclude that instead of fostering the emancipation of smallholders, development actors promote a labor-intensive and unprofitable way of farming that exploits local resources for the sake of green agendas and white markets. This article highlights the need for a critical reflection on the potential limitations of agroecology and calls for a more nuanced approach that considers the complex realities of smallholders in West Africa.

**Keywords:** agroecology; decolonial political ecologies; alternative food network; migration; climate change; Senegal





## 1. Introduction

We are writing the month of March 2019 and we are in a village in the Southern Niayes coastal region, not far from the Atlantic Ocean. We observe two friends, Ibrahim and Amadou, walking towards the sea in search of a better life. They have packed their belongings with the intention to set off in a pirogue in the direction of Europe. On their way, they cross paths with a man who understands what they are striving for. He tries to dissuade them from their plan and explains how dangerous the journey on those rickety boats was. He advises them to stay in Senegal and devote themselves to agroecological farming. Having listened to the man, Amadou decides to stay in his village and to engage in agroecology as the man had suggested. Ibrahim, however, sticks to his plan and embarks on his journey to Europe. Some years later, Ibrahim will return from Europe empty-handed. He will meet his old friend Amadou on a prosperous agroecological farm, producing healthy food for his wife and many kids and achieving a decent standard of living.

This story was staged as a theatre play on a square in a little village in the Southern Niayes coastal area. A part of our research team was doing exploratory fieldwork and was told about "a theatre piece about agroecology", by an agroecological smallholder that we had visited that day, and curiously we decided to go. Many residents from the village were attending—children, elderly, teenagers, women, men—everyone was there. The moral of the story was quite easy to understand for all of them: "Youth of Senegal, do not migrate! If you stay in the country and practice agroecology, prosperity will come your way!" Interestingly, the play was written and organised by a development agency from the Global North. After the show, a local partner of the organization presented the

benefits of agroecology to the community and announced the organization's new program that will support four people in the village with solar pumps for irrigation if they commit to agroecological farming.

Drawing from this field observation, the objective of this article is to examine the implementation of agroecology in Senegal and shed light on the curious case of development agencies organizing plays and the local implications that result from it. Currently, the promotion of agroecology is one of the top priorities of governmental and non-governmental organizations working in the areas of rural development [1]. Agroecology is commonly defined as an ecological alternative to industrial agriculture [2] and can be considered a scientific discipline, a set of agricultural practices, and a social or political movement [3]. Authors from different disciplinary backgrounds and development practitioners relate agroecology to a series of benefits that supposedly allow addressing some of today's biggest environmental, social, and economic challenges and, therefore, help achieve the sustainable development goals (SDGs) of the United Nations [1,4,5]. First, agroecological practices supposedly amount to climate change adaptation strategies: by eschewing chemical inputs and alternatively relying on the recycling of nutrients, enhancing soil organic matter, diversification of species, and integration of livestock, agroecology allows sustainable management of natural resources, enhances biodiversity and increases soil water retention [6,7]. This minimizes crop failure during droughts and makes the land more resilient to droughts and floods, which are more likely to occur as they are climate change induced [8,9].

Second, agroecology has the potential to improve working conditions and to foster job creation in agriculture. The absence of chemical fertilizers and herbicides results in health benefits for the agricultural workers, and the environmentally friendly farming style allows for reconciling ecological values with daily work and, therefore, increases the work satisfaction of farmers [3,10–12]. Further, agroecological techniques, such as crop rotation, polycultures, natural pest control, and compost making [7], increase the labor input compared to conventional high-input farming [13–18]. Therefore, some consider agroecological farming as an appropriate means to fight high unemployment rates through the creation of agricultural jobs [19,20], especially in so-called developing countries [21], where it is seen to slow the rural–urban exodus and international migration into Europe [9,22].

Third, it is argued that agroecology fosters the autonomy and sovereignty of farmers and food systems. Agroecology's integrated and holistic approach (comprising changes in production practices, knowledge co-production, and social and economic relations) and the integration of social movements has the potential to create more sustainable, just, and sovereign food systems [2,5,19,23–25]. In recent years, agroecology has been associated with various innovations related to alternative food networks, such as Participatory Guarantee Systems (PGS). They are said to enhance the sustainability of food systems and possibly empower local actors and foster local knowledge and short-value chains [26–28]. Although agroecological farming is often more labor intensive than conventional farming and usually produces lower yields [13,29,30], Altieri and Toledo argue that this paradigm has the potential for an "agroecological revolution" in Latin America because it fosters local self-reliance, the production of healthy food, and empowerment of peasant organizations vis-à-vis neoliberal agribusinesses [31]. Similarly, in the case of France, Coolsaet [32] stresses the counter-hegemonic and emancipatory potential of agroecology that may help farmers reclaim their autonomy. Yet others consider agroecology as a means to counteract capitalism's exploitation of human labor, and women's work in particular [33].

These positive views, however, are not uncontested. While some dub them as romantic optimism and peasant populism [34,35], others feel that agroecology traps African farmers in poverty due to inevitable productivity trade-offs rather than enhancing their independence [36]. Insights from various case studies around the world report poor working conditions [37], higher workload absorbed by unpaid family labor [38,39], unpaid labor arrangements such as internships [40,41], and attest agroecology (self-)exploitative tendencies due to heavy workload [42–44]. The considerable workload—the drudgery of work—has

been repeatedly identified as a barrier to adopting agroecological practices [45,46], especially for people engaged in care work [47–49].

Although the promotion of agroecology gains momentum in Sub-Saharan Africa due to the development nexus [50], empirical evidence regarding the organization of agroecological initiatives and their emancipatory potential for farmers remains scarce. Adopting a phenomenological approach [51,52], this article aims to address this gap through the case study of an alternative agroecological food network, a so-called PGS. We examine the web of actors and the discourses that feed into its creation and investigate related embodied experiences of smallholders and the potential for their empowerment. Exploratory fieldwork revealed pronounced North–South dynamics and smallholders' difficulties of implementation, leading to the hypothesis that the agroecological program reinforces the hegemonic position of the promoters of agroecology (development actors) but limits the empowerment of smallholders. This hypothesis was tested through knowledge co-produced during anthropological fieldwork. Informed by decolonial theories and feminist political ecology, the findings reveal aspects of coloniality of the PGS, which manifest in white spaces of alternative food networks [53], discourses of climate change adaptation as a means of preventing migration [54,55], and the reproduction of colonial tropes when explaining failures of implementation [56].

Based on these insights, we argue that the emancipatory potential of agroecology does not materialize for smallholders in Senegal due to externally defined terms that do not acknowledge local configurations. Conversely, the examined agroecology initiative appears to align with the history of foreign control of local resources for the sake of foreign (contemporarily green) interests. Drawing on Fassin's [57] notion of "humanitarian reason", we speak of "ecological reason" in the agroecological approachesthat have little consideration for local ecological knowledge and practices of production as well as of the organization of work.

## 2. Decolonial Political Ecologies

Scholars of post-modernism, postcolonial, and decolonial studies have been criticizing the "Western" notion of development for many decades [58–60]. Some scholars identified so-called development discourses that present countries, regions, or groups of people as traditional, isolated, or underdeveloped and prepare the ground for programs based on technical solutions that will result in modernization, market integration, or simply "development" [61,62]. These discourses, however, are rooted in hegemonic epistemologies and ontologies of the Global North [63,64] and may misinterpret and depoliticize the lives and needs of so-called target groups [62]. Those misrepresentations produce unintended and often harmful side effects and are ultimately responsible for the failure of uncountable development projects that create the need for new development projects [61].

It is argued that the structural power of transnational capital, which implies the ability to influence and shape institutions in their own interest [65], extends to development organizations and funding. This structural power is used for the diffusion of Northern concepts and categories in the South [60,66] and therewith exerts control over the postcolonial through operations of state and non-state organizations [67], even though involved actors may really have "the will to improve" the condition of the people [61]. Hence, although "colonialism ended with independence, coloniality is a model of power that continues" [68] (p. 229) through hierarchical power relations inherent to capitalism, neocolonialism, and international development [67]. This is mirrored in many interactions across the North–South divide [69]—including the ones embedded in the SDGs—and manifests, among other topics, in discourses surrounding climate change and in technical "solutions" to fight and mitigate climate change without mentioning power constellations and uneven pollution impacts [70,71]. While the so-called "developed countries" have contributed the most to climate change through their industries based on fossil fuels and high-consumption lifestyles, the "less developed" countries are responsible for significantly less $CO_2$ emissions but experience the climate change-induced effects most [72]. "Climate

coloniality" refers to the continuation of this uneven relationship through extractive industries, large-scale land acquisitions, conservation projects, REDD+ programs, but also through environmental development interventions, where actors from the Global North take advantage of resources and labor from the Global South [73–75]. Such development interventions can be analyzed through the lens of (bio)political ecology of climate change [55], which shows how global environmental discourses link the occurrence of migration to the effects of climate change. Hence, the assumption that climate change deprives people of their livelihoods and forces them to migrate [76] can result in the deduction of policy descriptions and development measures against climate change as a biopolitical technology aiming at containing migration [54]. According to Tacoli [77], migration can be an adaptive strategy to environmental changes but also to non-environmental factors. In many cases, mobility is not primarily connected to climate change but rather a consequence of measures that intend to promote adaptation and mitigation [74,77,78]. Against this background, the aim of decolonial studies is in line with the aim of feminist political ecologists that advocate for postcolonial intersectionality in order to uncover colonial practices and racial stereotypes embedded in these development practices [79]. The critical examination of mainstream development policies for the mitigation of environmental degradation entails the analysis of inherent patriarchal and northern ontological power and the interrelatedness with embodied realities of the "target people" [80,81].

Considering the above, we propose a decolonial perspective for analyzing alternative food systems that have emerged from North–South developmental relations. This perspective may seem counterintuitive at first, given that alternative food networks such as participatory guarantee systems have their roots in the originally decolonial idea of food sovereignty, which was shaped by leftist Latin American social movements such as La Via Campesina in their struggle against colonial domination and the neoliberalization of food production and consumption [35]. However, critical scholars have rightly drawn attention to the whiteness of spaces, bodies, markets, and discourses of many alternative agrifood movements, including organic farming, slow food, local grocery stores, farmers markets, and fair trade [53,82–84]. They understand whiteness as a socially constructed and not a biologically determined marker of difference [85], referring to a "structural advantage, standpoint, and set of historical and cultural practices" [86] (p. 81). Authors researching alternative food networks in the US write about the transportation of white dreams in the imagination of farming and rurality [87,88] that shape landscapes of labor and consumption [53,89–91], and they identify patterns of colonialism within them [82]. Situating her research in the US, Slocum [92] sees whiteness manifesting itself in the inequalities of wealth (among other things) that translates into the (in)ability to consume products of alternative food economies that are often produced by the labor of non-white immigrants. She rightly claims that this is valid for alternative food networks in other parts of the world, which we will see in the example of the PGS in Senegal.

This theoretical outline leads to the formulation of the following hypothesis: given that agroecology is promoted by development actors from the North, we assume that they have the power to shape these programs according to their own interests and use hegemonic discourses and narratives to justify it. We argue that by doing so, they misrecognize epistemologies of the South in general and local configurations and labor institutions in particular. This may entail unintended effects and lead to the failure of the projects for which the deficiency of local people is held responsible. Simultaneously, this situation creates the need for new environmentally friendly projects that reproduce the Northern development industry and primarily serve donor agencies' political interests rather than local smallholders' interests.

## 3. Field Site and Methodologies

The interdisciplinary research team AGROWORK is interested in factors enabling and hindering an agroecological transition in Senegal on different scales. For familiarization with agroecological initiatives and related actors, discourses, and narratives, the team

conducted expert interviews with representatives from NGOs, governmental development agencies, umbrella organizations, and farmer unions. It also visited different agroecological projects in the Thies and Diourbel in Western Senegal region that were identified as "agroecological" by either donor organizations or implementing partners. The PGS was chosen as a case study because it was the alternative food network that was most frequently mentioned and "advertised" during interviews with development organizations in the exploratory phase of fieldwork.[1]

A community administratively belonging to the city of Kayar (Figure 1) was chosen as a field site for a relatively high number of certified smallholders and for access to the field through prior established contacts. The main author conducted two blocks of ethnographic fieldwork from March to May 2019 and from February to June 2020 (interrupted by the coronavirus lockdown) and made a visit in April 2022 for the discussion and dissemination of research results. During the fieldwork, she was deeply immersed by living with two families where several members engaged in agroecological farming and enrolled in the PGS. Her participation in agricultural and domestic work activities in and beyond the families helped explore the local implementation of agroecological farming, challenges related to the organization of labor, rhythms of everyday life, interactions with donors, promoters, and technicians, and marketization.

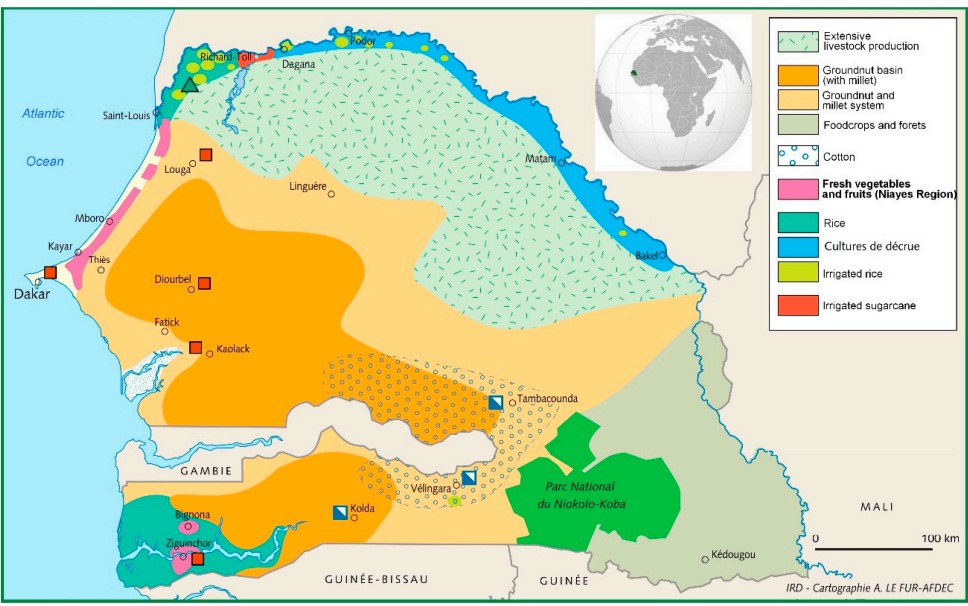

**Figure 1.** Map of Senegal with the Niayes area indicated by the pink color on the Western coastline. Source: IRD.

The exploratory fieldwork consisting of participant observation/observant participation [93], narrative walks [94], and open and semi-structured interviews [95] allowed for the identification of important themes and adjustment of the research questions. During the second block of fieldwork, they were operationalized more systematically through interviews, participatory photography [96,97], participant observation, and writing field notes and observation protocols [93]. In total, 110 interviews and informal conversations on labor in the larger sense were conducted with 65 people, most of them at least partially engaged in agroecology and 18 certified by PGS. A further 18 interviews with 12 representatives of NGOs and farmer unions engaged in the promotion of agroecology were realized. The length of the interviews and conversations was adjusted to the circumstances and availability of the research participants and ranged between half an hour and two hours. Some of them were recorded and others not, depending on the consent of the interlocutor(s), the context of the conversation, relations between researcher and interlocutor, and the formal/informal arrangement of the interview. In addition, numerous situations in every-

day life (spontaneous encounters, meals, common work activities) provided information relevant to the research and were documented in the research diary.

The co-produced findings were systematically analyzed using the coding and memoing technique based on the grounded theory approach [98] with the qualitative content analysis program MAXQDA. In the triangulation process, different sources are tested against one another, and reflexivity is applied to increase ethnographic validity [86,99]. The interpretation of data, especially the challenges related to the PGS, was discussed in a focus group discussion with research participants and bilateral discussions with research participants who could not participate in the focus group discussion.

We emphasize that the co-produced knowledge is situated [100] and does not constitute an objective truth. The adopted feminist approach [101] entails reflecting positionalities shaped by variables such as nationality, race, education, and wealth, which shaped interactions between research participants (mostly members of local communities partly engaged in agriculture, with low or middle-level of education and low or middle income), researcher (white outsider woman with higher education), and interpreters (a man from a local community, engaged in agriculture with a certain position in a farmer union, and a woman from the nearby town with a higher education and non-agricultural background). Power relations and asymmetries inherent in these configurations were mitigated to a certain extent through a longer-term stay of the first author in the community, her accommodation in local families and especially her participation in different working activities, which created cohesion and relationships of trust and respect.

## 4. (Post) Colonial Dynamics in Senegalese Agrarian Regimes

The colonial expansion of cash crops such as cotton in the Ivory Coast [102], cocoa in Ghana, tobacco in Nigeria [103], and groundnuts in Senegal [104–106] induced major socio-economic transformations and profoundly altered production relations [107]. The history of capitalist development of agriculture in West Africa is intimately linked to the formation of labor migration and would have hardly been possible without the transnational circulation of the additional workforce [104,107–111]. In Senegal, the agricultural expansion shifted to the peanut basin around the town of Kaolack when groundnuts replaced the gum trade as the main driver of economic activity [112]. Seasonal migrants called nawetan, originating from Mali, Guinea, and Burkina Faso, spend the rainy season as agricultural laborers in the Senegalese groundnut fields extending along the railway lines of the peanut basin [107]. At the turn of the 20th century, the French Colonial Administration introduced market gardening in the Niayes area and cultivated new crops such as cabbage, tomatoes, potatoes, onions, carrots, and eggplants. This sector would develop significantly during the droughts of the 1970s and 1980s for reasons connected to the availability of water and labor migrants from other parts of the country [113–116].

Since independence, agricultural policies in Senegal have been governed by different development plans. Until the mid-80s, the postcolonial state subsidized agricultural inputs heavily and facilitated marketization through government institutions. With the structural adjustment programs imposed by the international monetary fund in the early 1990s, the state disengaged completely and made way for the liberalization of the agricultural and food sector [117]. With the start of the new millennium, the Senegalese government implemented several agricultural programs designed to increase the national food production, to develop rural infrastructure, and also to prevent migration from rural areas to urban hubs and out of the country. Those programs were and are often financed by European states and cooperation agencies and became increasingly linked to anti-migration policies. Among them was REVA (Retour Vers l'Agriculture—Back to Agriculture) in 2006, which was financed for a large part by the Spanish government and designed to fight the migratory pressure toward Europe [118]. PRACAS (Programme d'Accélération de la Cadence de l'Agriculture Sénégalaise), starting in 2014, was mainly funded by foreign cooperation agencies [119], and ANIDA farms, also heavily supported by Spain, promised to "transform the rural exodus to an urban exodus" [120]. These programs are situated in a framework of

productivist agriculture that favors intensive and industrially based agriculture to increase productivity. It must be noted, however, that the Senegalese government has recently initiated a modest form of subsidization for imported organic fertilizers. This can be read as a response to the calls for a greener agriculture of supranational organizations and agroecological coalitions. This action has garnered criticism from advocates of agroecology who perceive it as a practice of input substitution instead of profound transformation of the farming system [121], and others who view it as a co-optation of the agroecological movement by government entities [122]. The Senegalese state, like other postcolonial states in the Global South, simultaneously supports both large-scale mechanized farming and small-scale family farming in its quest for development. The state tries to be responsive to different donor agencies, both the proponents of large-scale agricultural modernization and others (mostly non-governmental), who believe that family-friendly smallholder farming prevents social disruption [123]. In what follows, the authors will focus on the latter and outline the greening of agricultural practices in Senegal by introducing the most important actors and fields of intervention.

*Development Actors and the Panacea of Agroecology*

At the time of increased consciousness for environmental problems and the foundations of green political parties in Western Europe, the first agroecological initiatives in Senegal were implemented upon the initiative of European NGOs. ENDA-PRONAT, Agrecol Afrique, and other organizations started informing about the risks of pesticide use and trained farmers in ecological agriculture, seed reproduction, and manure making (expert interviews February and March 2019). At the beginning of the new millennium, agroecology in Senegal witnessed what Bottazzi and Boillat [124] call a "phase of proliferation" that manifested in the foundation of the Tafaé (TAFAE—Task force multi-acteurs pour la promotion de l'agroécologie au Sénégal), a platform of European NGOs in Senegal for the promotion of agroecology [125] and a growing number of organic and agroecological projects[2]. Furthermore, the establishment of FENAB—the national umbrella organization for organic and agroecological agriculture—with the support of German and Swiss funding is among the key developments of this phase. The member organizations of FENAB include NGOs and farmer unions that engage in capacity building and marketization of agroecological products through alternative food networks, such as the PGS, which will be analyzed in more detail below.

During a symposium on agroecology in 2015, the Food and Agricultural Organization of the United Nations declared Senegal a pilot country for agroecological farming in the West African region [9], and since then, agroecological initiatives have mushroomed in the country. Coordination and cooperation among the different actors increased and translated into the establishment of new networks and platforms such as DyTAES (Dynamique pour une Transition Agroécologique au Sénégal), a network uniting NGOs, farmer unions, rural women, research institutions, civil society organizations, and a network of local officials and businesses, which pursues the goal of fostering agroecology [126]. Those coalitions led to a partial formalization [124,127] of the efforts of the NGOs, which are backed by the internationally recognized FAO.

Although the actors have different foci in their agroecological programs (farmer training, marketization, lobbying) and different target groups (individual smallholders, women, economic interest organizations), they all depict agroecological farming as a panacea to fight climate change, strengthen local food systems, and local livelihoods [5,126,128,129]. The topic of climate change is omnipresent and easily justified by the fact that it manifests itself in Senegal visibly through processes of desertification, increased temperatures, salinization of water, irregular precipitation patterns, and the reduced availability of fresh water for human consumption and irrigation [115,130]. Development actors argue that "teaching" agroecology to smallholders enables them to adapt to climate change, enhance local food systems, and improve livelihoods and, in consequence, prevents them from migrating to the wealthier countries of the donor organizations. The link between devel-

opment programs and foreign anti-migration policies does not only express in reports of international organizations [22], national cooperation agencies [131], and conversations with representatives of NGOs but also manifests quite clearly in theatre pieces like the one described at the beginning of this article or on T-Shirts distributed to young men displaying slogans such as Rester ici, travailler ici, reussir ici (stay here, work here, succeed here) (Figure 2). These discourses also reflect in the design and implementation of the PGS, which aims to guarantee healthy products to consumers and claims to empower producers.

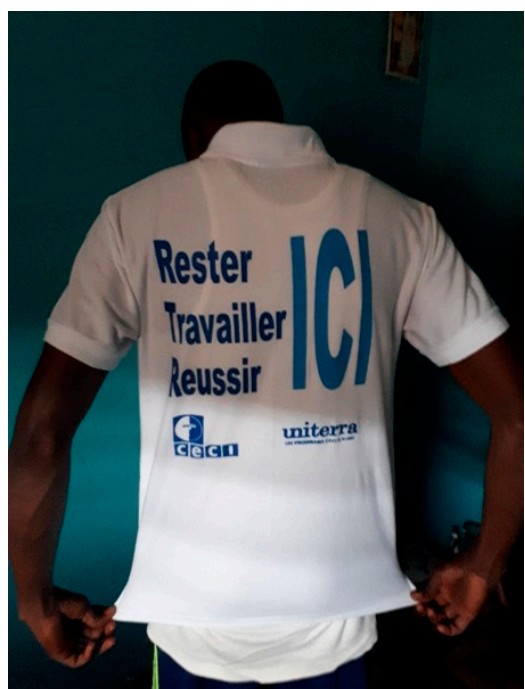

**Figure 2.** Young man with a T-Shirt distributed by an NGO.

## 5. The Participatory Guarantee System

The PGS is an alternative food system created upon the initiative of a Swiss NGO that had previously initiated the establishment of the National Federation for Organic Agriculture (FENAB). It was created with the idea of providing an affordable alternative to the French third-party certification Ecocert, which is too costly and too bureaucratic for many smallholders in the region and only has minimal effects on income (President of a farmer's union, 9 June 2020). In contrast with Ecocert, which certifies agricultural products for the European market, the PGS certifies vegetables and fruits for the domestic Senegalese market and therewith supports local value chains. The certification is based on the standard protocol, which was elaborated in a complex process involving international, national, and regional development institutions and farmer unions. Although PGS are said to be citizen-driven processes based on the active participation of all the relevant actors and stakeholders [132], we show elsewhere [133] that in the case of a PGS, international and national development actors had a hegemonic position in setting the rules of the game: FENAB and the International Federation of Organic Agriculture Movements (IFOAM) had the lead for the elaboration of the PGS on the base of a template of IFOAM, which sets the standards for PGS worldwide. The standard protocol that was accepted by IFOAM after almost a decade of negotiations stipulates the ban of chemical inputs and sets standards regarding intercropping, compost-making, crop associations, tree nurseries, seed reproduction and fabrication of natural pest control and defines the distances between agroecological and conventional plots, etc. Child labor is forbidden, but children above 12 years are permitted to work up to five hours per week in the form of training [134], but sharecropping, a widespread labor arrangement in the region, has been banned informally[3].

Once the body of rules was set up, the project started with two cycles (2017–2022) in the Niayes region, one of the five intervention zones of FENAB. Together with its member organizations, FENAB chose the Niayes region as a "pilot phase" due to its geographical proximity to Dakar, where an agroecological market was supposedly easiest to make for the relatively high purchasing power of (a part of) its population. Another key factor is the location of the offices of most of the involved NGOs in the capital city and nearby Thies, which enable the representatives and technicians to "move easily between their offices and the intervention zone" as the ex-president of an involved farmers union explained (interview 18 May 2020). Through the member organizations, FENAB identified 500 agroecological farmers that wanted to enroll in the PGS and get the label for their products. A certification committee—composed of representatives of the donor organization, FENAB, the four national NGOs supporting FENAB, and the consumers association—undertakes two controls per year, one announced and one unannounced. The committee controls the performance of the enrolled farmers according to the standard protocol and, depending on its observations, grants the certification for one year or denies or withdraws it in case of non-compliance with the rules. The certified farmers get stickers of the label and put them on the bags with the harvested products, which allows for marketization on three agroecological niche markets in Dakar. The certification guarantees a chemical-free and healthy product to the consumer and additional compensation for the efforts of a "clean production" to the producer [135]. The next subchapter will address issues of labor and markets by starting off identifying the people who are buying products in these markets and the people who are laboring to produce for these markets.

*5.1. Labor and Market Implications of the PGS*

Enrolled in the PGS are agroecological farmers from the Niayes area, which is the most important zone for horticulture, accounting for 80% of the production of the country. However, arable land is under pressure due to several reasons: Population growth and urbanization reflect in brisk construction activities, and climate change manifests itself through the salinization of water and the lowering of groundwater levels, which severely impacts the availability of the key resources of land and water [136]. Further, the area has been heavily degraded in the last 30 years due to excessive use of chemical inputs in agriculture [137], activities of agribusinesses and the extractive industry (cement, zircon, and phosphate) [138], and unmanaged dumping of domestic and industrial waste [139]. Land tenure insecurity is also endangering agricultural activities as customary law has been superseded by statuary law: with the law, no. 64-46, most land has become a national domain and can be reclaimed by the state for use in the public interest [140]. In the Niayes region, this has happened repeatedly for the realization of infrastructure projects or as concessions for extractive industries [138] and exposes the area to processes of "resilience grabbing" [141]. Today, the region is characterized by what Oya [142] terms "occupational multiplicity", meaning that the bulk of its residents engages in different income-generating strategies combining work in the sectors of agriculture, construction, petty trade, transport system, fishing, etc.; especially the sector of agriculture (both family farming and agrobusinesses) and fishing heavily rely on those "fluid labor migrations" [142] between the coast and the interior of the country. International migration into Europe is also part of the diversified income sources and is an adaptive strategy that contributes to the resilience of households [77,78,113,143][4].

The farmers enrolled in the PGS have been trained by their respective farmer union in different agroecological practices and have transitioned (or are still transitioning) from conventional to agroecological horticulture. They are supported by technicians of the farmer unions (who were trained by FENAB), who occasionally come to their fields to advise and correct them. However, most of the farmers report a series of challenges in meeting the requirements of the standard protocol.

First, they report higher labor intensity compared to conventional farming due to the absence of chemical herbicides that increase the need for tedious weeding. Second,

access to manure is a major challenge in the region as cattle rearing and farming activities tend to divide along the lines of different social groups [145], and this hampers collaboration between farmers and pastoralists. Moreover, smallholders' surfaces are often too small for cattle rearing, as plots are fragmented due to traditional inheritance rights or under pressure from large-scale investments [138,140]. Using someone else's manure, however, requires organization and payment for transport and investment of time. Third, many agroecological smallholders also mention challenges with intercropping because the required seeds are often unavailable (especially organic ones), and the right timing for nursing the different seedlings is hard to achieve under given circumstances. Fourth, the production ofnatural pest control with garlic, chili, tobacco, or the leaves of the neem tree (Azadirachta indica) is extremely demanding in terms of knowledge, labor, and time, and many farmers also consider it inefficient in case of pest attacks. Consequently, many experience crop failures and decimated yields due to vermin and fungi attacks. Finally, the informal ban on sharecropping additionally complicates labor implications, as the overall increased labor input of agroecological farming is accompanied by a decreased availability of additional labor power due to the sharecropping ban. Hiring labor by paying a salary is unaffordable for many women who do not have the necessary capital before harvesting, but it is also a challenge for many men who combine multiple jobs to assume the responsibility of providing for their families. The increased labor input conflicts with domestic and care work of women and multiple income strategies of men and women and further complicates distributional outcomes of agroecological horticulture on the grounds of class and gender [133].

Those who manage to cope with those difficulties and manage to respect the standard protocol of the PGS when cultivating their vegetables and fruits are entitled to sell their produce on what Boillat and Bottazzi [124] call agroecological niche markets in the chic neighborhoods in Dakar. Given that the certified products cost approximately 30% more than comparable conventional products (which are already perceived as expensive by many), exclusively wealthy people buy from this market. The clientele consists of white expatriates working for international development agencies, embassies, or research institutions and affluent Senegalese nationals, mostly working for the same institutions. These markets are perceived as white spaces [53,92] by agroecological farmers who also read the Senegalese costumers as white "because they live like the toubab [white people]" (agroecological smallholder, 8 March 2020). This means that these people live a white lifestyle for their standard of living, their consumption patterns, and their comfortable office work. This contrasts considerably with the lifestyle of the bulk of "ordinary local people" [146], who are mostly informally engaged in various occupational activities in the domains of agriculture and petty trade. For these reasons, we argue that those agroecological niche markets in Dakar and the occasional farmers' markets along the touristy Petite Côte coastline southeast of the capital constitute inherently white and thus racialized spaces.

### 5.2. Putting the Blame on Farmers: Reproducing Colonial Tropes

The difficulties in implementing the standard protocol for the PGS certification result in a high dropout rate of the program. In the beginning of the PGS project, 500 farmers enrolled in the program and aspired for certification. At the time fieldwork was completed, 135 out of the 500 had fully transitioned to agroecology, and 67 were certified. According to FENAB, the others are still in transition and will be certified soon. However, conversations with some of the allegedly enrolled farmers showed that many have dropped out of the program due to the challenges related to the standard protocol described above and due to market-related difficulties. Many agroecological farmers explicitly identify the combination of increased workload, lower yields, and lacking market as a decisive element for "being discouraged", as this farmer says: "They [the NGOs in their training] impose organic agriculture on us, but for fathers of a family, this is not good, there is no market. It is a lot of work and less yields. If you have a family, you must farm conventionally, otherwise

you won't eat" (smallholder in his 30s, married, father of five children, switched back to conventional agriculture and works as a postman and plumber).

This statement emphasizes the problem of the demand: The white niche markets are indeed only capable of absorbing a small number of certified products. Certified smallholders report having to sell the bulk of their products on conventional markets for conventional prices, which is economically unviable. For this reason, many smallholders switch back to conventional horticulture to increase cash income and meet the daily needs of their families. This citation illustrates that in in this context and under given circumstances, agroecology can lead to the opposite of the proponents' declared goals.

Representatives of promoting agencies explain these difficulties with problematic narratives that classify as stigmatizing. Confronted with the problems of low yields, missing market, and low income, a representative of an NGO gives his perspective on the question:

"These farmers are lazy. They know the good practices of agroecology, but you know what their problem is? They do not put the necessary effort, which means looking for organic fertilizers, composting, working the land properly, and assisting the field at all times; all these are the practices, but they say it is tedious, it is hard; they like it easy, that's it! Preparing and spreading agroecological fertilizer takes time, it's different from chemical fertilizer that you buy and spread straight away. The preparation of natural pesticides, organic phytosanitary products is slower, but less expensive" (interview, 22 February 2019).

The discourse of another NGO worker and representative of a farmer union fuels that trope of the lazy farmer who wants to earn money without working properly. He recalls how the certification committee caught farmers "cheating" when pest infested their plantation and says

"People do not have that patience. They prefer to cheat their way out of it, instead of being patient. One must do that effort. There's garlic treatment, neem leaves, they're very effective treatments. You have the products in your field, you can save money, but you need to make an effort with your children, do it yourself. It's work but it's not such hard work. It's no hard work" (interview, 18 May 2020).

This quote also transports a range of information that illustrates the gap between the views of NGO workers who are mostly unfamiliar with the realities of local farming and the embodied experiences of agroecological smallholders: The NGO representative suggests that using natural pest control saves money. This is a statement that is repeated by almost all the development actors but contested by manysmallholders. The latter explain that chemical pesticides are highly subsidized by the government and affordable for many. In contrast, the fabrication of natural pest control requires a lot of time and physical effort and conflicts with other types of work, such as domestic and care work. A female smallholder explains that the time she can spend in the field is limited, as she has other responsibilities such as domestic chores, children, and ceremonies she must attend (19 April 2019). A man also says that he would be in need of an additional labor force if he wanted to perform the agroecological practices correctly "because I have other activities too" (23 May 2020). Spending time in the field and fabricating natural pest control (besides weeding, manure making, watering, nursing seedlings, etc.) conflicts with other income-generating activities and means losing money if the other activities are compromised due to the time-intensive agricultural practices. Further, by stating that agroecology is 'no hard work', the interlocutor claims the power to define hard work and simultaneously delegitimizes embodied experiences and epis-temologies of smallholders.

## 6. Discussion: How Green Agendas Lead to Black Labourers for White Markets

This article has addressed the driving forces and the organization of agroecology in Senegal and scrutinized the emancipatory potential for local smallholders through the case study of the PGS in the Niayes region in Western Senegal. This important horticultural production zone is under pressure by processes of urbanization, environmental degradation, and land tenure insecurity caused by (post)colonial land and agricultural policies. In the course of its younger history, the Senegalese state has embraced differ-

ent agricultural policies, ranging from active engagement after independence to almost complete disengagement during the time of structural adjustment programs [107,117]. Since the beginning of the new millennium, we can observe the implementation of various programs following the logic of agricultural modernization that are often (co-)financed by European states, and that became increasingly linked to European anti-migration politics [118]. With the emergence of environmental values in the Global North, European NGOs started promoting organic and agroecological initiatives for smallholders in Senegal, trying to mitigate social disruptions caused by large-scale agriculture [123]. After the FAO had declared Senegal a pilot country for the agroecological transition in 2015, a veritable agroecological offensive started taking place. Currently, numerous NGOs, governmental development bodies, and supra-national organizations implement agroecological projects and discursively present agroecology as a panacea for fostering climate change adaptation, promoting food sovereignty, and enhancing local livelihoods, short, contributing to the achievement of several Sustainable Development Goals [1,4]. These expected outcomes are quite openly linked to the aim of reducing international migration [22], which is said to be caused by the effects of climate change that deprives people of the basis of their agricultural livelihoods [54,55,76]. Through programs consisting mainly of top-down technical training and advising, the promoters of agroecology aim to enable local smallholders—in reality, a differentiated group engaged in multiple occupations—to adapt to the effects of climate change, produce and sell healthy food and therewith uplift their overall living standard. This, so the assumption goes, will provide people with an improved perspective for the future and would incentivize them to stay where they are [5,9,22,126].

The establishment of the PGS embeds in this developmentalist discursive setting, with an overarching focus on climate change adaptation and food sovereignty. Development actors depict it as a locally anchored alternative to costly third-party certification. Given that agroecology favors short-value chains, the certified products are destined for the domestic market. The program is financed by a European NGO, and the standard protocol for certification was elaborated by the national umbrella organization (funded by European NGOs and cooperation agencies) and validated by IFOAM. The protocol defines the rules for certification, which range from the ban of synthetical inputs to detailed descriptions of agroecological practices [133]. Enrolled farmers are supervised by technicians who teach them the "good practices" and become certified by a team of different actors under the lead of the national umbrella organization. However, the extremely time and labor-intensive agroecological farming that would require full-time dedication to the farm collides with occupational multiplicity [142] that shapes most people's lives in the region. Against this background, many farmers report major difficulties when it comes to the implementation of the standard protocol. These difficulties arise through a combination of local biophysical and socio-political factors, and labor and time shortages on the one hand, and through unfavorable market configurations on the other. Comparatively few smallholders manage to conform to the standard protocol and are entitled to sell their produces on an agroecological niche market in Dakar for a price that is about 30% higher than for conventional produces. Given that these markets are too small and therefore unable to absorb the certified products, the agroecological smallholders sell most of their products on the conventional market for conventional prices, which is economically unviable and—combined with the other difficulties—results in a high dropout rate from the program.

Informed by decolonial political ecologies [67,82], the analysis of the co-produced knowledge unveils colonial patterns in the processes inherent in the PGS and detects elements of coloniality in the originally counter-hegemonic idea of agroecology [147] on different levels. Starting with the local and moving on to the global, they describe as follows: First, the alternative food network established through the PGS manifests in a racialized landscape of producer–consumer relations. The niche markets established through the PGS, appear to be white spaces [53,83–85,89–92], where exclusively European and North American expatriates, tourists, and wealthy Senegalese nationals can afford to do their shopping. The Senegalese customers are equally read as white [146] by the

agroecological smallholders for their linkages to international organizations, capital, and values, and their high consumption lifestyles and therefore insert into the whiteness of the market. However, given that only a limited number of people have white purchasing power, the market is unable to absorb the certified products that were produced under the constraints of additional labor requirements. Concretely, this means that black bodies are talked into a green and labor-intensive, and economically unviable way of farming through hegemonic developmentalist discourses while only white bodies can consume a guaranteed pesticide-free and environmentally friendly product.

Second, depicting smallholders as being too lazy to correctly implement the "good practices" amounts to the reproduction of old colonial and stigmatizing tropes. It ignores the daily reality of many people that shape through a combination of various occupations that leave little time for the implementation of time-intensive agroecological practices that go at the expense of other income-generating activities and social reproduction. In this way, the responsibility for the difficulties and failures of the agroecological projects is put on the smallholders and helps disguising the developmentalist misreading of local realities [62]. These discursive practices relieve development agents from the responsibility of critically re-thinking the universality of their values and approaches and emphasize their belief in disposing of the "good solutions" that only fail due to the insufficiency of others. This reinforces asymmetric power relations in favor of the Northern actors [63,64] and weakens the position of the less powerful groups and discourses [67].

Third, by linking international migration to the effects of climate change, development actors prepare the ground for foreign policy interventions [76] in the form of environmental sustainable development that responds to the interests of European countries to contain in-migration through the prism of green values and the fight against climate change [54,55,70]. However, this discourse obscures the fact that international migration is an adaptive strategy not only to climate change but also to non-environmental factors [77,78] that contributes to the economic resilience of households [113,143] and is therefore perceived largely positive in Senegal [144].

Fourth, the narratives of fighting climate change and providing just food systems are embedded in the paradigms of the SDGs and, therefore, resonate with donor agencies. This allows attracting funding for respective programs and allows the development sector consisting of supra-national organizations, national cooperation agencies, and NGOs, to reproduce itself [61]. A possible conclusion is that the main beneficiary of these programs is not the so-called "target group", hence local smallholders, but rather aid workers from the Global North and a part of the African elite working for the same organizations or local offshoots [127].

Based on these insights, the authors argue that the emancipatory potential of agroecology does not unfold in Senegal because the respective initiatives—which also contain positive elements—are embedded in asymmetric power relations rooted in the colonial past and in discourses and assumptions that are coined by Northern epistemologies and ontologies. The technical top-down approach of the PGS does not change colonially coined power asymmetries and land and agricultural policies, and insufficiently takes into account local socio-ecological and socio-political realities and market relations. Contrarily to the claim of enhanced autonomy, agroecology in the given case rather seems to amount to a green anti-politics machine than a counter-hegemonic way of farming [62,70]. Drawing on the co-produced knowledge analyzed in the article, the PGS seems to insert itself in the long history of foreign control of local labor and resources for the sake of foreign interests. In the case of the agroecological PGS this manifests in a green agenda and white markets, and foreign policies shaping biopolitics in a black and formerly colonized region.

In order to make agroecology an emancipatory project, the involved actors would need to move away from purely technical solutions and take into account the wider dynamics of a given context, local resilience patterns (including international migration), power relations, local values, organization of farm, off-farm and reproductive work and given market relations. Future research may provide systematically researched socio-demographic

profiles of research participants in order to overcome the limitations of this study. However, defining categories such as "main occupation" and "family" and understanding the complexity of collecting information about income (given multiple income sources but also the circularity of money within extended families and social networks) is methodologically and ethically very challenging and needs to be done carefully [148–151]. However, it might be worthwhile to do the exercise in order to generate more precise knowledge about different groups of local people and the respective potential to be (dis)empowered. Future research should also explore locally feasible ways of more sustainable farming practices that acknowledge local realities and market and labor relations, including migrant labor. These studies should adopt a strictly participatory and transdisciplinary approach and conceptualize local smallholders as a differentiated group of people that simultaneously engage in many other income-generating and time-consuming activities. For the Senegalese government, one possible step could be to increase subsidies for organic inputs and improve smallholders' access to them. This input substitution would not contribute to the desired emancipatory project of convinced agroecologists [10,31,121] but could be an initial step towards more environmental sustainability in Senegalese horticulture that avoids the current detrimental trade-offs with social and economic sustainability. It could reduce labor intensity, allow for multiple accumulation strategies and possibly reduce prices that would make healthy products accessible to a non-white group of consumers too.

**Author Contributions:** Conceptualization, F.M.; fieldwork, F.M. and P.B.; writing—original draft preparation, F.M.; writing—review and editing, T.H. and P.B.; funding acquisition, P.B. All authors have read and agreed to the published version of the manuscript.

**Funding:** This research was funded by the Swiss National Foundation for Scientific Research (SNSF), AgroWork project grant number 176736.

**Data Availability Statement:** Not applicable.

**Acknowledgments:** The authors would like to thank the editors for their efforts in organizing the special issue and selecting the article for publication. The first author would like to express her gratefulness to all her research participants for sharing their knowledge and experiences. She would further like to thank the interpreters for their assistance and the different farmer unions for providing access to information and members. Special appreciation goes to the hosts for accommodation and generous support.

**Conflicts of Interest:** The authors declare no conflict of interest.

## Notes

[1] There are other alternative value chains established by foreign NGOs, but they are characterized through a different form of organization (cooperative or economic interest organizations). Given that they are embedded in the same socio-economic context, they face similar challenges as the PGS.

[2] The definitions of organic and agroecological agriculture often diverge in literature and practice. While organic agriculture makes use of organic fertilizers and pesticides, agroecology normally rejects external inputs and relies on practices such as intercropping, crop diversification, and manuring for pest control. However, promoters and, smallholder in Senegal do not clearly distinguish the two and often use these notions synonymously, thereby referring generically to more sustainable practices that exclude the use of chemical fertilizers and herbicides.

[3] Sharecropping is a labor arrangement in which a plot owner grants access to his/her land to a person who does not have access to land (often labor migrants from the interior of the country that either do not have land or have land and cannot cultivate it due to the absence of rain and irrigation system during the dry season). The sharecropper normally receives shelter, food, inputs, and land from the landowner, and he cultivates that land during one agricultural cycle. The harvest is sold, and the added value is split equally between the landowner and the sharecropper. In the frame of the PGs, sharecropping is banned informally because executive members of the leading non-governmental organisations were afraid sharecroppers would use chemicals to boost the yields that are their payment; hence sharecropping is banned for the sake of protecting the purity of the products.

[4] International migration is commonly linked to social (upwards) mobility reflected in the saying: "Tukki, Tekki, Tedd, Teral" which translates into "travelling, making it, succeeding socially, and helping family and friends" Sall 2011 cited in [116] and is therefore quite positively perceived in the Senegalese society. The perspective of accumulating relative wealth abroad and

being able to accommodate the family upon return shapes the dreams and aspirations of many young men and reflects in many conversations in everyday life [144].

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
