# Peer review of "Green Agendas and White Markets: The Coloniality of Agroecology in Senegal"

_land, doi:10.3390/land12071324_

Round 1

Reviewer 1 Report

Comment for the authors:

First of all, I congratulate the authors of the article for the research and the fieldwork they have carried out, since the text presented for evaluation is well founded, structured, and developed. In addition, they study a very relevant social and political reality of agroecological practices in Senegal. It is very relevant the studying of this country, declared in 2015 a pilot area for agroecological farming in the West African region.  However, it is necessary to clarify and / or develop some ideas.

1.- In research articles it is necessary to emphasize the central ideas and questions of the research carried out. In this article, the working hypotheses are not explicitly indicated. It is necessary to indicate the hypothesis or hypotheses that are worked on in the article. This can help readers identify the central ideas under investigation. It would also be useful to make explicit how the hypotheses are linked to the article's research question. However, as this article presents an ethnographic and qualitative research, perhaps the hypotheses emerge after analyzing the discourses of the interviewed actors. If this is the case, the hypotheses that arise from the data provided by the actors studied must be made explicit. Concretely, the grounded theory approach (page 5) allows the formulation of hypotheses in this last form.

2.- I suggest to the authors of the article that, to the extent that this is possible, they expose the socio-demographic profiles (age, gender, educational level, main occupation, and income level, and number of members of the family of the farmer) of all the actors studied in the article. This information, which can be very useful to interpret the main points of the article, can be presented in a Table in the Methodology section.

3.- In the Discussion, it is convenient to indicate the limitations of the research carried out. In other words, it is recommended that the final epigraph of the article explicitly comment on the possible limitations of the research presented in the article. This can add more rigor to the research presented and give the authors new ideas to continue their work in the future. For example, one of the limitations of the research presented is that it does not take into account socio-demographic profile of all the actors studied.

4.- In the final section of the article (Discussion), it is convenient to show the new research questions (and possible hypothesis) that arise after the study carried out. The latter allows continuing the research undertaken in the near future, by the authors of the article or other researchers.

Author Response

Please see response letter attached.

Reviewer 2 Report

1. In the section ”Decolonial Political Ecologies”, please provide some logic diagrams or structure diagrams to make it easier for the reader to understand the logical relationships.

2. In the section “Field site and methodologies”, display the survey area and samples on the map to give readers a more intuitive understanding of the research area.

3. At the end of the article, please suggest implications for other African countries and what should be done to respond.

Author Response

Please see response letter attached.
